# Pulse Burst Generation and Diffraction with Spatial Light Modulators for Dynamic Ultrafast Laser Materials Processing

**DOI:** 10.3390/ma15249059

**Published:** 2022-12-18

**Authors:** Zheng Fang, Tong Zhou, Walter Perrie, Matthew Bilton, Jörg Schille, Udo Löschner, Stuart Edwardson, Geoff Dearden

**Affiliations:** 1Laser Group, School of Engineering, University of Liverpool, Brownlow Street, Liverpool L69 3GH, UK; 2College of Science, Jiangsu University of Science and Technology, Zhenjiang 212100, China; 3SEM Shared Research Facility, University of Liverpool, Liverpool L69 3GH, UK; 4Laserinstitut Hochschule Mittweida, University of Applied Sciences, Technikumplatz 17, 09468 Mittweida, Germany

**Keywords:** ultrafast, materials micro-structuring, spatial light modulators, pulse burst processing, laser induced periodic surface structures (LIPSS)

## Abstract

A pulse burst optical system has been developed, able to alter an energetic, ultrafast 10 ps, 5 kHz output pulse train to 323 MHz intra-burst frequency at the fundamental 5 kHz repetition rate. An optical delay line consisting of a beam-splitting polariser cube, mirrors, and waveplates transforms a high-energy pulse into a pulse burst, circulating around the delay line. Interestingly, the reflected first pulse and subsequent pulses from the delay line have orthogonal linear polarisations. This fact allows independent modulation of these pulses using two-phase-only Spatial Light Modulators (SLM) when their directors are also aligned orthogonally. With hybrid Computer Generated Holograms (CGH) addressed to the SLMs, we demonstrate simultaneous multi-spot periodic surface micro-structuring on stainless steel with orthogonal linear polarisations and cylindrical vector (CV) beams with Radial and Azimuthal polarisations. Burst processing produces a major change in resulting surface texture due to plasma absorption on the nanosecond time scale; hence the ablation rates on stainless steel with pulse bursts are always lower than 5 kHz processing. By synchronising the scan motion and CGH application, we show simultaneous independent multi-beam real-time processing with pulse bursts having orthogonal linear polarisations. This novel technique extends the flexibility of parallel beam surface micro-structuring with adaptive optics.

## 1. Introduction

Ultrashort, femtosecond and picosecond laser pulses are ideally suited for the study of linear light–matter interactions, for example, during pump–probe spectroscopy [1], non-linear laser lithography for large area nano-structuring [2], and for industrial applications such as selective ablation for thin film solar cells [3,4], micro and nanoscale surface modification for control of chemistry and wettability [5], generation of superhydrophobic and antibacterial surfaces [6], and creation of complex Laser Induced Periodic Surface Structures (LIPSS) [7,8]. For Industrial applications involving micro-machining of deep-milled structures, volume ablation rates need to be high (δV/δt > 3 mm^3^/min), so kHz, low power, multi-Watt ultrafast laser systems are simply not adequate. This limitation has been addressed recently by scaling ultrafast laser output to the multi-hundred Watt level, achieved primarily by increasing repetition rates from tens of MHz to GHz [9,10], although kHz, ultrafast, high energy thin disc systems with pulse energy E > 200 mJ, 1 kW average power and pulse length τ < 1 ps have been developed [11]. The flexibility of high-power ultrafast laser technology has also been expanded by introducing burst modes at much higher intraburst frequencies (GHz-THz) [12], which alter the laser–material interactions, leading, for example, to improved surface finish. Burst mode ablation can improve micro-structuring by introducing so-called ablation-cooled material removal [13]; although the material-dependent ablation efficiency with pulse bursts can be limited to low pulse numbers per burst [14] or even display lower efficiency with detrimental thermal effects and plasma shielding, similar to effects observed with nanosecond laser pulses [15,16,17]. In the case of metals, an up-to-date review of ultrafast laser ablation with double pulse and pulse bursts has been recently carried out [10].

Adaptive optics, such as Liquid Crystal on Silicon Spatial Light Modulators (LCOS SLM), can increase the flexibility of light use when combined with ultrafast lasers, accelerating laser micro-fabrication throughput via parallel processing [18,19]. These devices can modulate phase, amplitude, or polarisation, allowing optimisation of the laser parameters on target while using most of the available output pulse energy since ultrafast laser micro-machining often requires only μJ energies/spot due to the low ablation thresholds [20,21].

Incident polarisation is a critical parameter in surface micro-structuring, and Hasegawa and Hayasaki [22] demonstrated material ablation with two orthogonal linear polarised spot arrays employing femtosecond pulses, 2 SLMs, and a high NA objective, achieving impressive nano-structuring (~140 nm) of an ITO film. Mikhaylov [23] used a combination of beam splitting with a delay line and 2 SLMs, allowing a two-pulse burst micro-machining study on stainless steel for an industrial stamping application. The 2D intensity uniformity was limited only by the proximity of identically polarised spots. Lou et al. [24] micro-structured silicon with arrays of defined femtosecond cylindrical vector (CV) beams. The simultaneous creation of up to 20 CV beams using a single digital Hologram on an SLM combined with interference has been well demonstrated [25]. SLM technology can also handle high peak (>50 GWcm^−2^) and average power (*p* > 200 W) exposures [26]; hence, a highly useful tool when combined with ultrafast laser sources, matching the optical landscape to the task at hand.

The motivation behind the current work is to demonstrate the ability to convert a low-frequency ultrafast laser system from the kHz range to >300 MHz pulse frequency, significantly extending the capability of a low average power ps laser system. The research work here introduces a simple pulse burst optical system able to split a relatively high energy (E_P_ > 100 μJ) ultrafast laser pulse into two or more pulses in an optical train with a few nanosecond temporal separation; hence, intraburst frequency ν_IB_ > 300 MHz at the fundamental 5 kHz frequency. The pulse bursts include pulse pairs with orthogonal linear polarisations and hence can be modulated independently with two SLMs. This additional degree of control allows a demonstration of dynamic, real-time, near-simultaneous inscription of linear and complex LIPSS with orthogonally polarised pulse bursts at ν_IB_ = 323 MHz.

This novel system, which combines a pulse burst generator with 2 SLMs, allows a direct comparison of the laser–material interactions between kHz repetition rates and multi-hundred MHz burst processing where intra-pulse delays are approximately ~3 ns.

## 2. Materials and Methods

### 2.1. Optical Setup

A schematic of the optical layout setup is shown in Figure 1. The laser source, an Nd:VAN seeded regenerative amplifier (High Q Laser, Rankweil, Austria, IC-355-800 ps), has output pulse duration τ = 10 ps, wavelength λ = 1064 nm, repetition rate 5 kHz, and beam diameter φ ~ 2.3 mm in a quasi- Gaussian spatial mode with linear, horizontal polarisation. The beam was attenuated and passed through a diffraction-limited beam expander (M = x3, Rodenstock, Munich, Germany), then entered the optical delay line (pulse burst generator), which consists of a half-wave plate (HWP_1_), polarising beam splitter (PBS), dielectric reflecting mirrors, and internal wave plate (HWP_2_) within the loop. The relative setting of the HWP_1,2_ fast axes creates reflected and circulating pulses or a pulse burst.

The burst pulses are then incident at low AOI < 10° on SLM_1_ addressed with appropriate CGH and re-imaged via a 4f system (f1 = f2 = 300 mm) to SLM_2_ where, after further modulation, the beam was again re-imaged (via 4f system, f3 = f4 = 400 mm) to the input aperture of a scanning galvo system (Nutfield) and focused with a 100 mm focal length f-theta lens to the substrate. The scan software employed was SCAPS GmbH. The beam profile of pulse bursts prior to materials micro-structuring could be observed using a flip mirror and imaged to a Spiricon camera (SP620U) by a long focal length lens (L_5_, f = 750 mm). The SLMs are reflective phase-only type, Hamamatsu X13138-5785 (1280 × 1024 pixels) and X10468-03 (800 × 600 pixels), respectively, and SLM_2_ was mounted with its director vertical, orthogonal to that of SLM_1_. The CGHs shown are simple hybrid holograms used for beam shaping [27] of the quasi-Gaussian laser beam (elliptical) to circular. The inner circular regions have a fixed phase, while the outer region with high spatial frequency diffracts away from the peripheral elliptical component. The first SLM_1_ modulates |H〉 polarised pulses from the delay line while SLM_2_ modulates only |V〉 polarised reflected pulses from the PBS. Pulse bursts of diffracted horizontal and vertical polarised arrays could be generated with appropriate hybrid CGHs while arrays of orthogonal vector beams [24] could be created by inserting a nano-structured S-wave plate (SWP) [28,29] just beyond (and in close proximity) to SLM_2_. Polished 304 stainless steel samples (roughness Ra = 50 nm) were mounted on a three-axis (x,y,z) motion control system (Aerotech) to bring the sample surface to the focal plane.

### 2.2. Pulse Burst Generator

Referring to Figure 2, HWP_1_ rotates the incident linear polarisation at the PBS interface; hence, the amplitude of reflected, |V〉 polarised and transmitted |H〉 polarised components are determined by the HWP_1_ fast axis setting. The internally transmitted pulse circulates around the delay line, and if no internal waveplate were added, it would exit the delay line at the PBS with |H〉 polarisation, thus creating an orthogonally polarised pulse pair. The addition of HWP_2_ within the delay line alters internal polarisation so that the circulating pulse is again split into transmitted |H〉 and reflected |V〉 components at the PBS. The relative setting of external and internal waveplates thus allows the creation of a pulse burst with burst number N_P_ ≥ 2 with polarisations, |V〉, |H〉, |H〉, |H〉….and so on. The optical path length from the PBS centre around the loop was measured to be ΔL = 91.7 ± 0.5 cm. Allowing for the PBS dimensions and its refractive index (2.5 cm cube, RI = 1.5), the effective optical path delay of the loop OPD = 93.0 ± 0.5 cm and hence the burst frequency ν_IB_ = 322.8 ± 1.6 MHz determined primarily by the speed of light in air. We designate pulse bursts as (1,1,0) for two pulse orthogonal polarisations with a 3.1 ns delay, while other pulse sequences can be designated (1,0,1) or (0,1,1) for 2-pulse burst orthogonal polarisations (6.2 ns delay) or identical |H〉 polarisations (3.1 ns delay) respectively. Similarly, (1,1,1) represents a 3-pulse burst with the two delayed pulses, |H〉 polarised, orthogonal to the first pulse |V〉 polarised.

A schematic diagram of a 3-pulse burst output at 5 kHz fundamental frequency with ν_IB_ = 323 MHz intra-burst frequency is shown in Figure 3.

## 3. Results

### 3.1. Hybrid CGHs

The use of a flip-up mirror with a long-focus lens (f = 750 mm) allowed the delay line burst output to be imaged to a Spiricon CCD camera to check relative spot intensities/spatial overlap prior to surface micro-machining experiments. Figure 4a shows a 3-pulse (1,1,1) CCD image from the delay line with nearly equal pulse energies, spatially separated using a slight misalignment of the delay line, then spatially overlapped, Figure 4b. No correction with hybrid CGHs are used here, and the result of blind drilling on stainless steel with the overlapped pulse burst is shown in the optical image, Figure 4c, with near elliptical cross-section and eccentricity ε = 0.72. The laser mode from the regenerative amplifier is only quasi-Gaussian and so eccentricity ε = ω_0_ (min)/ω_0_ (max) where ω_0_ (min) and ω_0_ (max) are the measured ablation semi-minor and semi-major axes, close to the 1/e^2^ radii at intensity focus. Figure 4d shows the effect of using hybrid holograms (Figure 1), which diffract away the outer intensity wings from the ellipse major axis. Figure 4e shows the overlap intensity profile, significantly improving beam roundness, while Figure 4f demonstrates (1,1,1) pulse burst processing with hybrid CGHs yielding eccentricity ε = 0.88, a great improvement but at the cost of a reduction in overall transmission efficiency.

Figure 5a,b show the hybrid CGHs used to create a 2 × 2 spot orthogonally linearly polarised Gaussian array, while Figure 5c shows the resulting CCD image of the 2 × 2 Gaussian array. By adding the nano-structured SWP to the optical line just after SLM_2_, the orthogonal linearly polarised Gaussians were converted to a vector beam array with orthogonal (Radial/Azimuthal) polarisations and ring intensity distributions, Figure 5d. One can see evidence of low-energy ghost beams here, expected from patterns with a high degree of symmetry when calculating inverse Fourier Transforms (IFTs) [30].

The high purity of the vector beam polarisations was confirmed by adding a wedged beam splitter (B/S) to the CCD camera while rotating the CCD by 90°. As the B/S angle of incidence (AOI) is close to the Brewster angle, it acts as a polarisation analyser. Vector beams with radial/azimuthal polarisations, |Ra〉, |Az〉 and their superpositions, |Ψ〉 = α|Ra〉 + βe^iδ^|Az〉 (where δ is a phase angle) are shown in Figure 6. The double lobe structure with an intensity null on the axis is expected from vector beam transmission through a linear polariser [7].

The double lobes are well-defined, indicating a high vector beam purity. These vector polarisation states can be represented on the equatorial axis of a high-order Poincare sphere (HOPS) while scalar ring modes with pure optical angular momentum OAM (e.g., m = ± 1, phase e^± iϕ^) and |R〉, |L〉 circular polarisations appear at the poles [31]. In between the poles and the equator, the states are cylindrical vector vortex (CVV) beams. The Laguerre–Gaussian, LG (0,1)* spiral phase mode can be represented by a superposition of two orthogonal degenerate Laguerre–Gaussian (LG) modes, given by [32],
(1)$$E\left(r,\varphi \right)=E_{0}\sqrt{\rho \;}\;e^{-\rho /2}\cos\varphi \;e_{x}+E_{0}\sqrt{\rho \;}\;\;e^{-\rho /2}\sin\varphi \;e_{y}\;$$
where ***e_x_***, ***e_y_*** are unit vectors along the *x*, *y* axes, *r* and *φ* are cylindrical coordinates where *ρ* = 2 *r*^2^/*ω*^2^ and *ω* is the 1/*e*^2^ radius of the Gaussian from which the LG mode can be generated. The intensity distribution reflected from the wedged plate (transmission axis along horizontal axis ***e_x_***) is then given by (***e_x_* ● *e_y_*** = 0)
(2)$$I=E^{*}\cdot E=I_{0}\rho e^{-\rho }\sin^{2}\varphi$$
which has a double lobe structure and intensity null at the centre due to the exponential term.

### 3.2. Fast Photo-Diode Outputs

Pulse burst amplitudes from the delay line were checked using a fast photo-diode (PD, Thorlabs DET025A/M, risetime τ ~ 100 ps) connected to a wide bandwidth digital oscilloscope (Tektronix 3054, 500 MHz, 5 GS/s). Figure 7 shows oscilloscope traces from output pulse bursts when adjusting the external and internal half-waveplate in the delay line. Figure 7a shows a (1,1,0) pulse pair with a 3.1 ns delay, while Figure 7b shows the (1,0,1) pulse pair with a 6.2 ns delay. Figure 7c shows the (1,1,1) 3-pulse burst at intra-burst frequency ν_IB_ = 323 MHz with equal amplitude. Figure 7d shows a 4-pulse burst with decaying amplitude. It was not possible to generate equal amplitude pulse bursts for N_p_ > 3. The temporal pulse shape does have a low-intensity component arriving approximately 2 ns before the main pulse. Surprisingly, adjustment of the Regen amplifier Pockels cell voltage and intracavity waveplate did not reduce this component appreciably, while the seed oscillator pulses showed no such temporal structure.

### 3.3. Ultrafast Ablation with Orthogonal Linear Polarisations

With equal energy (1,1,0) pulse bursts, orthogonally linearly polarised and spatially diffracted, the resulting surface structuring observed on polished stainless steel with a 2 × 2 Gaussian spot array is shown in the SEM (JEOL 7001FEGSEM) images of Figure 8a–c. Pulse exposure/spot was N_p_ = 100 with pulse energy/spot E ~ 3 μJ (F_0_ ~ 0.3 Jcm^−2^); hence, total burst energy on target E_p_ ~ 12 μJ. While Figure 8b shows the complete symmetric ablation pattern, Figure 8a,c with higher magnification exhibit low-frequency orthogonal LIPSS with pitch Λ ~ 1 μm at the outer edge of the exposed regions.

### 3.4. Pulse Burst Ablation with Orthogonal Radial and Azimuthal Polarisations

With the addition of the cylindrically birefringent SWP to the optical line, orthogonal linear polarisations were converted to ring mode Radial and Azimuthal polarisations. This depends only on the direction of the linear polarisations relative to the direction to the SWP axis. Thus, two incident orthogonal linear polarisations from a modulated pulse pair created simultaneous Radial and Azimuthal polarisations. Figure 9a shows SEM images of a 2 × 2 array micro-machined simultaneously on 304 stainless steel with orthogonal Radial and Azimuthal polarisations. Figure 9b shows the higher magnification images of the resulting orthogonal complex LIPPS which are well-defined, supporting the view that the vector polarisation states are quite pure (Figure 5). Exposure pulse number N_p_ = 100/spot with total pulse energy E ~ 7 μJ/spot (F_0_ ~ 0.26 Jcm^−2^) and average power <*P*> = 140 mW exposure at the material surface. The ring mode peak fluence = (1/e) F_0_ (Gaussian).

### 3.5. Scanning with Orthogonal Linear Polarisations, Fixed CGHs Polarisations

LIPSS are quasi-periodic surface undulations whose source can be regarded as due to the superposition of the incoming wave with a surface scattered wave which affects the spatial energy deposition on the surface [33,34]. The key parameters for producing LIPSS are the temporal pulse length (fs-ps), laser wavelength, fluence, pulse overlap, pulse exposure, and linear polarisation direction relative to scan direction. There is, therefore, a relatively wide processing window over which LIPPS can be created; however, overexposure leads to a reduction in the finesse of the LIPSS. On s. steel, a fluence in the range F = 0.3–0.4 Jcm^−2^ was found to be effective with a 10 ps pulse length.

Galvo scanning on the surface with orthogonal polarisations allowed fast surface LIPSS texturing. Figure 10a,b shows optical images of WL illumination of a diffractive chess board, micromachined simultaneously at 5 kHz with orthogonal linear polarisations from the 2-pulse (1,1,0) burst with Gaussian beams, inscribing LIPSS approximately at right angles when scanned in the x direction. The spot separation was adjusted to 1 mm, galvo scan speed was s = 40 mm/s, pulse overlap ~33%, and hatch distance d = 20 μm. Pulse energy/spot E = 3 μJ (F_0_ = 0.3 Jcm^−2^). Illumination directions are indicated with red arrows. The optical image in Figure 10d shows that surface texturing is highly polarisation-dependent, while Figure 10c,e show SEM images of the LIPSS formation, which are not orthogonal. This is due to an **E** field rotation within the Galvo and LIPSS formation is sensitive to scan direction. When the **E** field vector is nearly parallel to scan direction, we observe stronger coupling so that low-frequency LIPSS form normal to the major **E_x_** component. On the other hand, with **E** tilted away from the y direction (**E_y_** major component) while scanning in x-direction, LIPSS form normal to the **E** field direction, not that of the **E_y_** vector. This can be resolved by rotating scan directions by approximately 20°.

### 3.6. Dynamic Polarisation Modulation and Micro-Structuring

As nematic liquid crystal SLMs can be addressed in real-time, the application of CGHs can be synchronised to the scan motion control system. Two series of CGHs were designed (lens and gratings algorithm) to demonstrate independent real-time modulation of a two-pulse burst (1,1,0) with near Gaussian spots. A TTL trigger signal from the laser shutter was sent to a NI input/output controller (NI USB 6501), which signalled a Labview environment to synchronously start the CGH application to the two SLMs. The shutter was opened for 20 ms (50 pulses) and closed for 980 ms every second, which matched the CGH frequency, set to 1.0 Hz. Hence, we demonstrate the synchronous application of CGHs to both SLMs while independently modulating their relative spatial separations during a pulse burst. Figure 11a,b shows microscope images of the resulting surface micro-structuring when CGHs modulate the (1,1,0) orthogonal two-pulse burst with sinusoidal functions, which have a π/2 phase shift while the stage was translated at v = 30 μm/s. The surface LIPSS are clearly orthogonal, as expected, Figure 11b.

### 3.7. Comparison of Burst Mode with Single Pulse Processing

Ablation characteristics of pulse burst processing compared to low-frequency (kHz-MHz) have been shown to vary significantly in the literature [15,16,17]. We confirm that this is indeed the case, firstly during blind drilling. Typical surface texture with 5 kHz exposure, N_p_ = 150 pulses, Ep = 4 μJ/pulse (F_0_ = 0.4 Jcm^−2^) is shown in the SEM image, Figure 12a, with clear low-frequency LIPSS and little evidence of material melting, expected at fluences a few times ablation threshold. On the other hand, 3-pulse burst (1,1,1) ablation at the intraburst frequency ν_IB_ = 323 MHz with the same fluence/pulse (N_p_ = 150 pulses) demonstrates a major difference in surface texture with evidence of melting and surface smoothing, Figure 12b. LIPSS (orthogonal to those with 5 kHz exposure) are still apparent in an outer ring but effectively disappear within a large region of the exposed area. This remarkable change in surface texturing is likely due to plasma absorption and heating by the |H〉 polarised pulses, coming within 3.1 ns delays, the time scale on which the plasma plume expands during ultrafast laser ablation [35,36,37], resulting in additional plasma surface heating [38,39].

Figure 13a,e show higher resolution SEM imaging of the 5 kHz pulse and pulse burst processing, respectively, while the corresponding EDX (Oxford Instruments Ultim-Max EDS detector with AZtec software) elemental analysis of the Oxygen Kα1 maps, (b) and (f) indicate a higher oxygen level at the periphery of the ablated spots. Re-deposition of Fe oxide nano-particulates occurs during the combustion of the excited Fe atoms in air, yielding Fe_2_O_3_ and Fe_3_O_4_, which have enthalpies of 824 kJ/mol and 1118 kJ/mol, respectively [40]. Other elemental maps for Fe Kα1, (c), (g) and Si Kα1 (d), (h) show no significant change in surface atomic composition. The acceleration voltage was 15 kV with electron penetration depth estimated to be 1–1.5 μm.

We machined stainless steel in a square pattern, 0.4 mm dimension and 20 μm deep with 5 kHz, (1,0,0) and (0,1,0) exposure, with 2-pulse bursts (1,0,1), (1,1,0) and 3-pulse bursts (1,1,1), respectively. Pulse fluence was F_0_ = 0.4 Jcm^−2^, scan speed s = 40 mm/s, and hatch distance = 20 μm. The residual surface micro-structure for 5 kHz (1,0,0) exposure shows clear LIPSS, Figure 14a, while with the (1,1,1) pulse burst, no sign of LIPSS appears, Figure 14b, a significant change in surface texture, typical of all pulse burst processing results.

Ablation rates were estimated from the micro-machined squares, 0.4 mm × 0.4 mm × 20 μm deep at fluence F = 0.4 Jcm^−2^. Ablation depth increased linearly with pulse number in every case, but we found volume ablation rates decreasing as follows: R(1,0,0) ~ R(0,1,0) > R(1,0,1) > R(1,1,0) > R(1,1,1) with ablation volume/pulse = 6.9 μm^3^/pulse, 6.7μm^3^/pulse, 4.6 μm^3^/pulse, 3.2 μm^3^/pulse, 2.7 μm^3^/pulse, respectively. The ablation rate of the (1,1,1) pulse burst is, therefore, only 39% of that at 5 kHz. Interestingly, the ablation volume/pulse of the (1,1,0) pulse burst (3.1 ns delay) is 30% lower than that of the (1,0,1) burst with a 6.2 ns delay. These observations strongly support plasma absorption as the source of the decreasing ablation rate with pulse bursts and are in agreement with the experimental results of other authors [14,15]. While ablation rates are lower with pulse burst processing, nevertheless, the surface finish improves due to plasma heating and melting of the surface.

## 4. Discussion

There is currently great interest in pulse burst processing of materials with ultrafast lasers [9]. Here, we converted a low repetition rate 5 kHz, 10 ps, ultrafast laser source to high-frequency pulse bursts at the fundamental frequency using a burst generator delay line consisting of a polarisation beam splitter with external and internal HWPs combined with 3 dielectric mirrors so that reflected and delayed pulses could be brought collinear. The relative setting of the two HWPs altered the energetic 10 ps laser pulses (f_0_ = 5 kHz) to a pulse burst with intra-burst frequency ν_IB_ = 323 MHz or 162 MHz. Up to 3 pulses with equal energy could be generated, allowing the study of the pulse burst laser–materials interactions and comparison with kHz processing. As the pulses reflected from the PBS were vertically |V〉 polarised while transmitted components were horizontally |H〉 polarised, this fact allowed independent polarisation modulation and diffraction of the pulse bursts with the aid of two phase-only SLMs whose directors were also orthogonally aligned.

By employing hybrid CGHs on each CGH, beam shaping of the elliptical laser mode was achieved, altering eccentricity from ε = 0.72 to ε = 0.88; hence, spot roundness was improved significantly, although at the cost of lower transmission efficiency. Consequently, pulse burst arrays of spots with orthogonal linear polarisations were modulated simultaneously, imprinting surface LIPSS with orthogonal directions. By introducing a nano-structured SWP into the optical line, CV beam arrays with ring modes and orthogonal polarisations (e.g., Radial/Azimuthal) simultaneously imprinted more complex low-frequency LIPSS, also confirming the high purity of these interesting polarisation states, regarded as classically non-separable in spatial and polarisation modes [41,42]. As nematic liquid crystal SLMs can be addressed in real-time, synchronisation of CGH application to the SLMs was demonstrated at 1 Hz in a Labview environment, allowing independent dynamic diffraction of the (1,1,0) orthogonally polarised pulse pairs.

A comparison of 5 kHz multi-pulse (1,0,0) and 3-pulse burst processing (1,1,1) at 323 MHz intraburst frequency on fixed spots demonstrated a remarkable change in surface texturing of stainless steel during blind drilling. While low-frequency LIPSS are characteristic of multi-pulse ultrafast kHz ablation at fluences just above the ablation threshold, pulse bursts with nanosecond delays introduced significant surface melting due to plasma absorption, which, using the pump–probe technique, can be maximum after a few ns delay [34]. With pulse bursts on stainless steel, ablation efficiency decreased with increasing pulse burst number; 2-pulse (1,1,0) ablation efficiency η = 46% while 3-pulse, (1,1,1) η = 39% of that measured at the fundamental 5 kHz frequency, respectively. This is strong evidence of plasma absorption due to the following pulses in a burst which arrive within 3.1 ns of each other. The residual surface texture with pulse burst processing appeared much smoother than 5 kHz, so burst processing could be regarded as a valuable finishing process to improve surface finish after a lower repetition rate kHz micro-machining of metals.

With an optical setup involving two SLMs, a 4f optical system, and a delay line burst generator, it is worth reporting the overall efficiency of laser light used at 5 kHz compared to a pulse burst with and without beam shaping. With the (1,0,0), 5 kHz output and no beam shaping, the overall transmission efficiency from the PBS to the Galvo input aperture was measured to be η = 63%, while with the hybrid CGHs applied to both SLMs, this dropped to η = 50%. With the (1,1,0) two-pulse burst without beam shaping, η = 61%, while with hybrid CGHs applied, efficiency dropped to η = 43%. These results compare favourably with those measured by Hasegawa and Hyasaki [22] with two SLMs who quoted η ~ 8% with an fs laser source and 2 SLMs. However, they also demonstrated up to 20 diffracted spots with orthogonal linear polarisations.

Very recently, a delay line consisting of a non-polarising beam splitter (40:60, 20:80) and three mirrors was used to create pulse bursts, all with the same polarisation at 1 GHz intra-burst frequency for fs laser processing of steel and copper [43]. Bursts with up to 5 pulses and decaying amplitudes could be generated, also showing a reduction in ablation rate on steel with burst mode. Domke et al. [16] used a linear delay line with a PBS, 2 mirrors, and quarter-wave plates (QWPs) to cleverly double their burst frequency from a femtosecond laser at 77 MHz to 154 MHz. Although not stated, a pulse burst with up to 28 pulses and decreasing amplitude had orthogonal polarisations from pulse to pulse. Fraggelakis et al. [44] demonstrated large area, 2D nano-structured LIPSS on stainless steel with fs double pulse cross polarised exposure using interpulse delays in the range 0.1 ps ≤ Δt ≤ 50 ps.

The work presented here also demonstrates dynamic polarisation and spatial modulation of orthogonally polarised pulse bursts with 2 reflective SLMs, increasing the flexibility of laser processing of materials. If the output laser beam were truly Gaussian, the losses incurred by beam shaping would be reduced, hence, increasing light use efficiency to η ~ 63%, although this will be CGH dependent. The technique developed in this work would allow simultaneous processing at kHz repetition rates combined with pulse burst processing, dependent only on CGH application, and might well be applied in the spatial control of surface materials wettability, surface bio-compatibility, or encoding complex surface structures for security marking of valuable components. In addition, it may be possible to directly write high-quality reflective optical gratings or plasmonic sensors into metallic surfaces. By optimising the synchronisation scheme, orthogonal burst polarisation modulation at frequencies >10 Hz should be possible, limited only by the SLM bandwidths <30 Hz [45]. With current cooled SLMs able to handle high average powers >100 W [26], then very high throughput large area pulse burst processing with orthogonal polarisations becomes possible.

## 5. Conclusions

An optical pulse burst optical system operating at intra pulse burst frequency of 323 MHz from a 5 kHz fundamental frequency has been developed to study the differences in ultrafast laser–material processing of stainless steel. At 5 kHz, micro-structuring produces clear low-frequency LIPSS both for fixed spots and during scanning when the beam overlap is approximately 33%. On metals, these low-frequency LIPSS, with period Λ ~ λ (laser wavelength), appear near right angles to the local E vector. On the other hand, pulse bursts with 3.1 ns between pulses lead to plasma absorption, which re-heats the surface, leading to melting, so that LIPSS disappear. As a two-pulse (1,1,0) burst has orthogonal polarisations, each pulse can be modulated independently by an SLM with directors oriented appropriately. This has allowed independent, real-time, two-beam surface processing with orthogonal and complex LIPSS. The current Galvo system with a 100 mm focal length f-theta lens has a 60 × 60 mm^2^ flat scan field, while the translation stages have a 100 mm range. By combining these motions, the processing area can be increased to approximately 130 × 130 mm^2^, industrially relevant. The novel optical system developed expands the current window utilising techniques available for high throughput, dynamic ultrafast laser processing, particularly when combined with kHz, high energy, high average power systems currently under development [11].

## Figures and Tables

**Figure 1 materials-15-09059-f001:**
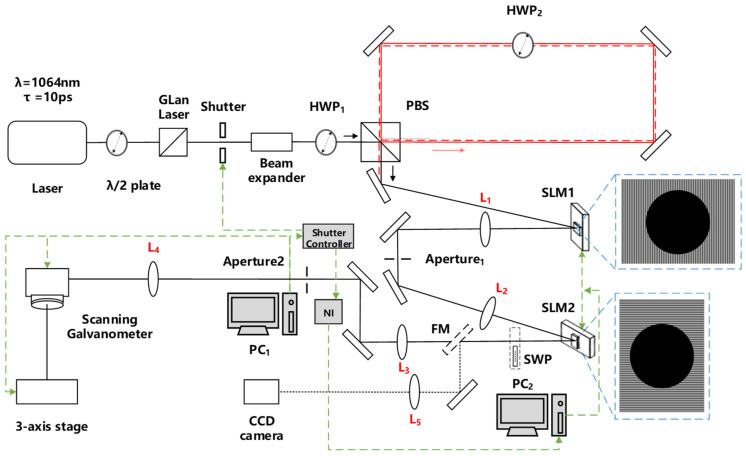
Schematic of the optical setup. The attenuated and expanded beam is directed to the delay line, HWP_1_/PBS plus mirrors, and HWP_2_. The pulse burst is controlled by altering the relative settings HWP_1_ and HWP_2_ (within the delay line). The resulting bursts are modulated by the SLMs with a 4f system (L_1_/L_2_, f = 300 mm) re-imaging diffracted spots to SLM_2_. A second 4f system (L_3_/L_4_, f = 400 mm) re-images all diffracted beams to the input aperture of scanning galvo. A flip mirror with a long focal length lens (L_5_, f = 750 mm) focuses resulting intensity distributions to a CCD camera. Samples are brought to the lens focal plane by a 3-axis stage. For dynamic control of CGH application, two PCs with a trigger signal from the shutter controller and NI input/output were employed.

**Figure 2 materials-15-09059-f002:**
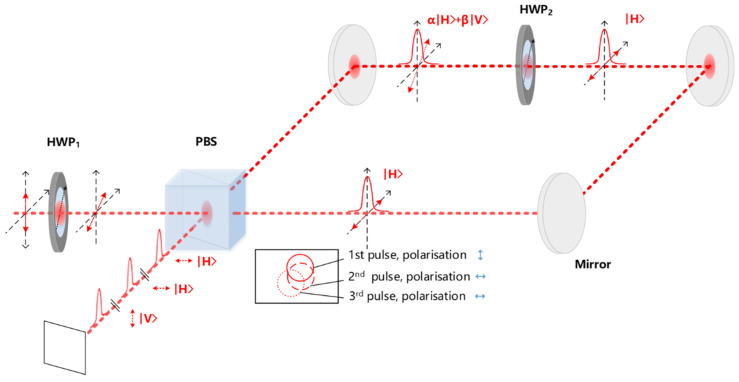
Schematic of the pulse burst generator setup with external HWP_1_, PBS, dielectric mirrors, and internal HWP_2_. An incident pulse linearly polarised with E field direction, θ to the horizontal axis is split at the PBS, creating reflected, |V〉 polarised and transmitted, |H〉 polarised components. The polarisation of the transmitted pulse is then rotated within the delay line by HWP_2_ splitting again at the PBS so generating the pulse burst. Polarisation vectors on reflected/transmitted beams are shown, while the setting of the fast axes of HWP_1_ and HWP_2_ determines the pulse burst characteristics. Careful adjustment of the delay line mirrors allowed spatial overlap of the pulse burst in both the near and far fields. Temporal pulse separation Δt = 3.1 ns.

**Figure 3 materials-15-09059-f003:**
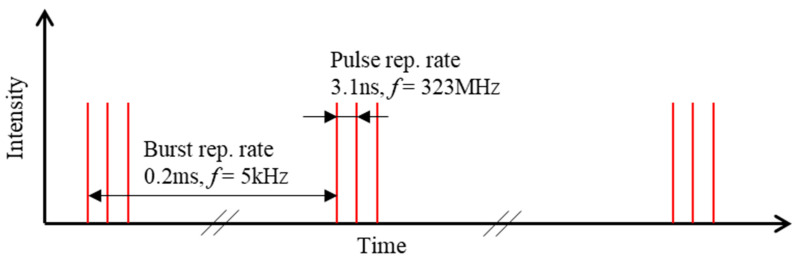
Schematic of laser burst mode from delay line with 3 equal intensity pulses. The fundamental repetition rate f_0_ = 5 kHz while the intra-pulse burst repetition rate f_B_ = 323 MHz.

**Figure 4 materials-15-09059-f004:**
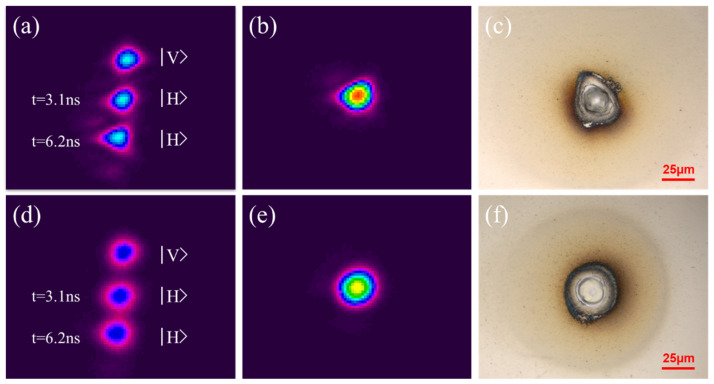
Blind drilling with pulse bursts (**a**) No hybrid CGHs, CCD image of spatially separated 3-pulse (1,1,1) burst with equal pulse energy, showing quasi-Gaussian mode, (**b**) CCD image with pulses spatially overlapped (**c**) blind drilling on stainless steel (E = 4 μJ, F_0_ = 0.4 Jcm^−2^, Np = 150) with overlapped pulse burst yielding spot ellipticity ε = 0.72. (**d**) hybrid CGHs applied, CCD image of 3-pulse (1,1,1) burst, spatially separated (**e**) pulse burst spatially overlapped demonstrating improved spot roundness (**f**) blind drilling on stainless steel with (1,1,1) pulse burst with hybrid CGHs yielding spot eccentricity ε = 0.88.

**Figure 5 materials-15-09059-f005:**
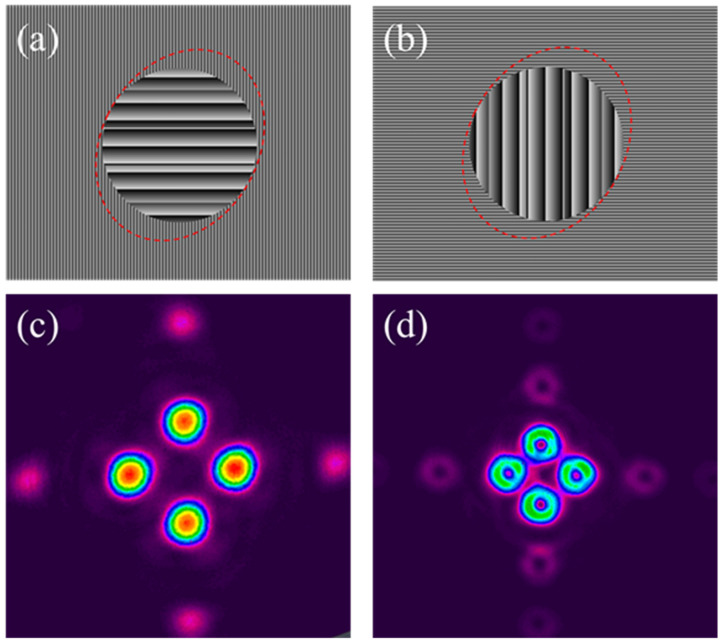
(**a**) Hybrid CGH on SLM^1^ generating 2 horizontally |H〉 polarised diffracted Gaussian spots (**b**) hybrid CGH on SLM_2_ generating 2 vertically |V〉 polarised Gaussian spots, (**c**) CCD image of the 2 × 2 Gaussian spot array with orthogonal linear polarisations, (**d**) vector beam array with Radial/Azimuthal polarisations resulting from the addition of the nano-structured SWP after SLM_2_.

**Figure 6 materials-15-09059-f006:**
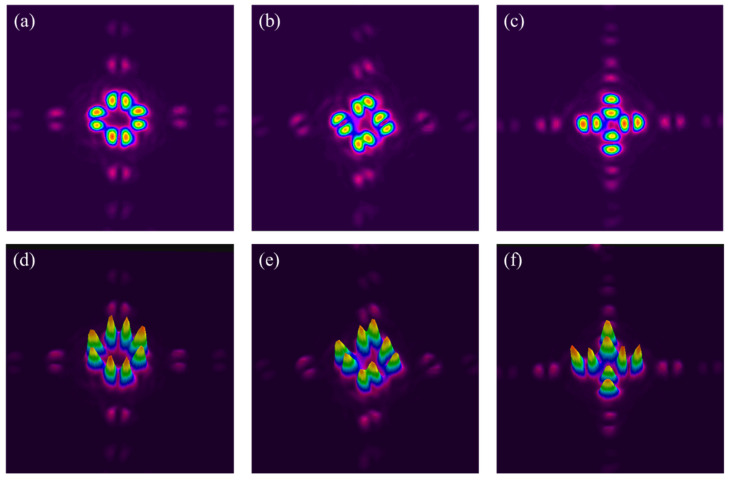
Polarisation analysis of simultaneous orthogonal vector beams from pulse burst (1,1,0) imaged on the CCD camera when rotating SWP axis (**a**) 0°, (**b**) 45°, (**c**) 90°. Their corresponding 3D intensity profiles are shown in (**d**–**f**), respectively. The profiles correspond to Radial and Azimuthal polarisations (**a**,**c**) and their superpositions (**b**,**e**).

**Figure 7 materials-15-09059-f007:**
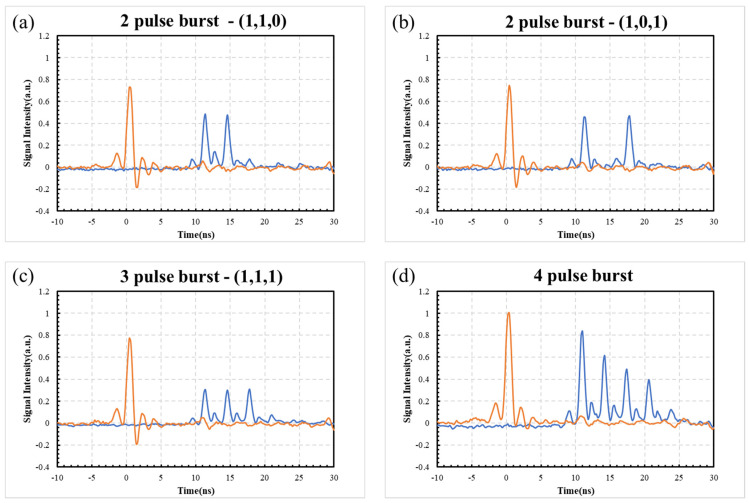
Waveforms of pulse bursts using fast PD/digital oscilloscope, (**a**) 2-pulse (1,1,0) burst, 3.1 ns delay, (**b**) 2-pulse burst (1,0,1), 6.2 ns delay, (**c**) 3-pulse burst (1,1,1), equal energy at intra-burst frequency = 323 MHz, (**d**) 4-pulse burst with decaying amplitudes, 323 MHz. The red trace represents the 5 kHz Regenerative amplifier output from a fast internal laser PD. There is evidence of a low-intensity component, 2 ns ahead of the main pulse from the Regenerative amplifier. This typically amounts to <15% in the pulse bursts.

**Figure 8 materials-15-09059-f008:**
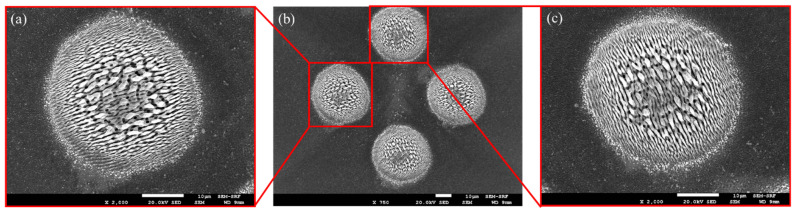
SEM images of simultaneous inscription of surface LIPSS on stainless steel with orthogonal polarisations (**a**) 2 × 2 spot array, *n* = 100 pulses, E = 3 μJ/pulse/spot, F_0_ = 0.3 Jcm^−2^ (**b**) higher magnification of micro-structured spot with horizontally |H〉 polarised exposure, (**c**) higher magnification of micro-structured spot with horizontally |V〉 polarised exposure.

**Figure 9 materials-15-09059-f009:**
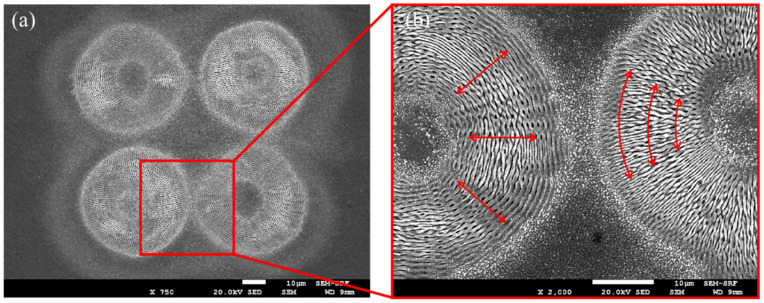
SEM images of simultaneous ablation with orthogonal cylindrical vector beams (**a**) 4 spot Radial and Azimuthal polarisations (**b**) higher magnification SEM image showing clear LIPSS generated by these remarkable polarisation states. The local E field vectors are shown in red.

**Figure 10 materials-15-09059-f010:**
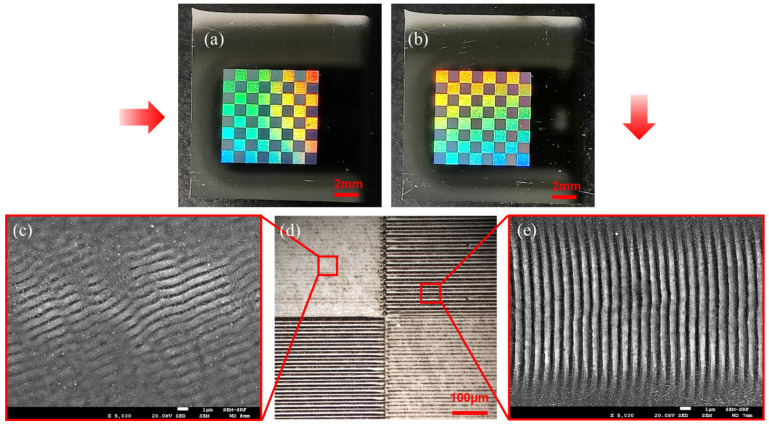
Two-pulse burst large-area processing with orthogonal linear polarisations to produce a diffractive chess board, 8 × 8 mm in size, due to periodic surface micro-structuring (**a**) WL diffraction from the chess board, 8 × 8 mm with 1 mm squares. Red arrows indicate illumination directions so that the dark squares have LIPSS direction approximately parallel to illumination direction, (**b**) orthogonal WL illumination, reversing the diffractive areas, (**c**) SEM image of clear LIPSS inscribed with E field approximately orthogonal to scan direction, (**d**) Optical microscope image of the intersection region of squares, (**e**) SEM image of LIPSS inscribed with E field parallel to scan direction. The small white scales on the SEM images have 1 μm length.

**Figure 11 materials-15-09059-f011:**
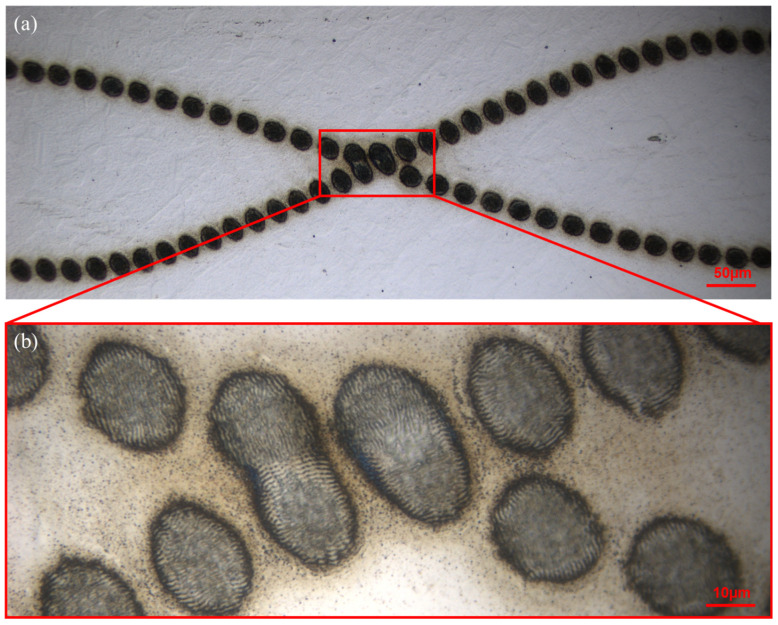
Dynamic pulse burst micro-structuring on stainless steel with real-time spatial modulation of two orthogonally linearly polarised spots (1,1,0) in a sinusoidal motion with a π/2 phase shift, (**a**) low magnification optical image, 50 pulses/spot, F_0_ = 0.3 Jcm^−2^ (**b**) high magnification optical image showing crossing point and orthogonal low-frequency LIPSS. Scan speed = 30 μm/sec. Note that spots here are elliptical as the creation of a large number (2 × 64) hybrid CGHs in Matlab required a significant additional effort.

**Figure 12 materials-15-09059-f012:**
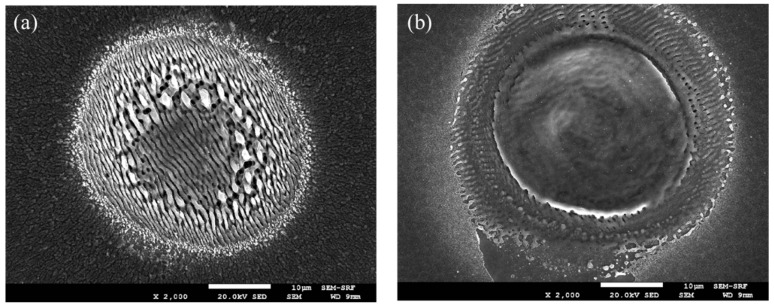
SEM imaging of multi-single pulse ablation versus three-pulse burst, (**a**) typical ablation at 5 kHz with N_p_ = 150 pulses, E = 4 μJ, F_0_ = 0.4 Jcm^−2^ (**b**) three-pulse (1,1,1) ablation (N_p_ = 150) with the same energy/pulse which shows a smooth texture in the centre region, likely due to surface melting from plasma absorption and heating. LIPSS orthogonal to those from kHz exposure (1,0,0) appear in an outer ring.

**Figure 13 materials-15-09059-f013:**
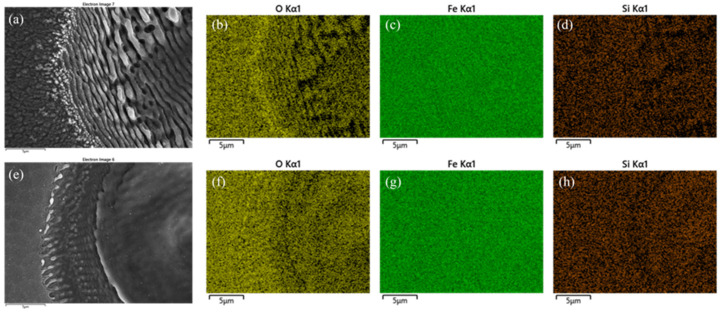
(**a**) SEM image of 5 kHz pulse ablation, F_0_ = 0.4 Jcm^−2^ N_p_ = 150 pulses, (**b**–**d**) EDX maps of surface ablation at 5 kHz showing increased Oxygen Kα_1_ signal at ablation edge, while Fe Kα_1_ and Si Kα_1_ show negligible change (**e**) burst processing (1,1,1) with same energy/pulse, N_p_ = 150 showing surface melting due to plasma absorption, (**f**–**h**) EDX maps of the Oxygen Kα_1_, Fe Kα_1_ and Si Kα_1_ signals for pulse burst with slightly increased Oxygen signal outside melted region while Fe Kα_1_ and Si Kα_1_ again show negligible change. The acceleration voltage was 15 kV with electron penetration depth estimated to be 1–1.5 μm.

**Figure 14 materials-15-09059-f014:**
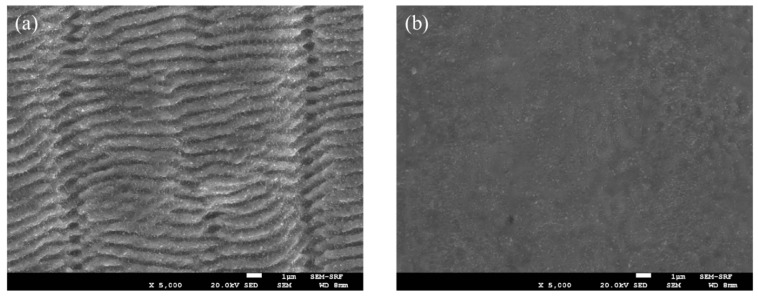
SEM images of residual surface micro-structure of 20 μm deep, 0.4 mm squares (**a**) (1,0,0) 5 kHz exposure with clear LIPSS, F_0_ = 0.4 Jcm^−2^ (**b**) (1,1,1) 3−pulse burst processing, F_0_ = 0.4 Jcm^−2^ with no LIPSS apparent. Scan speed s = 40 mm/s, with pulse overlap ~30% and scan line offset = 20 μm.

## Data Availability

The data presented in this study are available on request from the corresponding author.

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
