# Peer review of "Pulse Burst Generation and Diffraction with Spatial Light Modulators for Dynamic Ultrafast Laser Materials Processing"

_materials, 2022, doi:10.3390/ma15249059_

Round 1

Reviewer 1 Report

This manuscript  uses a delay line with polarizing elements inside to create bursts of ultrafast laser pulses (fs) spaced by ns. This gives control on laser polarization, with first pulse polarized orthogonal to the following ones. Three was the maximum number of pulses of equal power, but larger number of pulses could be obtained with decreasing power. The polarization and inter-pulse interval control give excelent flexibility in structuring stainless steel surface ablation. The use of two perpendicular Spatial Light Modulators (SLM) with convenient focal arrangements increased malleability in laser beam control. The manuscript show excelent examples of different surface textures.

Even though the manuscript has more evolution in laser optics and laser beam manipulation, the important application in surface material texturing was very well demonstrated  and it is convenient for publication in Materials.

Reviewer 2 Report

This paper reports on the fabrication of LIPSS. It is technically interesting. However, there is the possibility of a fundamental misunderstanding. Also, some points should be explained for the reader's understanding.

>In the text, LIPSS means Laser Induced Plasmonic Surface Structures, but the correct meaning is Laser Induced Periodic Surface Structure. Wording correction and general content correction is required.

p.5 3.1

>Please explain the relationship between "e" and intensity.

>Please explain the cause of the asymmetry shown in Figs. 4(a) and 4(b).

>Please clarify the time order of the beams shown in Fig. 4(a) and 4(d).

>Please explain why the spread around the beam (brown area) shown in Fig. 4(f) is larger than that in 4(c).

>The dark images in Figs.5(c) and 5(d) are described as ghosts, but they are also possible diffraction patterns. Please explain which is the correct understanding on this.

>Please explain whether the beam splitter HWP has anti-reflection coating.

>From the photographs shown in Figs. 10(a) and 10(b), we can see that this LIPSS is not caused by plasmons, but by the effect of the diffraction grating. In this paper, the color of LIPSS is generally discussed as plasmon, but it is a color due to diffraction, so it is desirable to correct all of them.

>Why are the individual patterns shown in Fig. 11 elliptical? Please add an explanation.

>Please specify the acceleration voltage and electron penetration depth in the EDX measurement shown in Fig.13. Please add information on other elements (such as Fe) in SUS steel.

P.14

>Please explain the relationship between the heat capacity of the sample and LIPSS in this experiment. Are the values discussed in the text optimized values? Please add an explanation including whether or not there is only one solution (value).

>Please add an explanation about the possibility of increasing the LIPSS area by the method of this paper.

Reviewer 3 Report

Comment:

The Authors Fang et al presented a manuscript with the title of “Pulse burst generation and diffraction with Spatial Light Modulators for dynamic ultrafast laser materials processing”. In this work, the authors reported a pulse burst optical system, able to alter an energetic, ultrafast output pulse train to 323MHz intra-burst frequency at the fundamental 5kHz repetition rate. The experimental work has been well planned and performed. This work may be interested to the researchers/readers in the related community. I recommend the publication of this manuscript (pending possible minor revision).

My comments are list as below.

1.       The motivation and novelty of this draft is not clearly presented.

2.       Fig. 1: The authors should spell GL in full.

3.       There is no conclusion section in this draft. The authors should have a conclusion section in this draft.

Overall, the experimental results in this work are impressive. The reviewer recommends publication of this work (with possible minor revision).

Round 2

Reviewer 2 Report

The manuscript has been revised according to the suggestions and comments of the reviewer.

Thank you for your contribution.

Others

In the conclusions section

60 x 60 mm -> 60 x 60 mm2

130 x 130 mm -> 130 x 130 mm2